# LOVECon: Text-driven Training-free Long Video Editing with ControlNet

## Abstract

Leveraging pre-trained conditional diffusion models for video editing without further tuning has gained increasing attention due to its promise in film production, advertising, etc. Yet, seminal works in this line fall short in generation length, temporal coherence, or fidelity to the source video. This paper aims to bridge the gap, establishing a simple and effective baseline for training-free diffusion model-based long video editing. As suggested by prior arts, we build the pipeline upon ControlNet, which excels at various image editing tasks based on text prompts. To break down the length constraints caused by limited computational memory, we split the long video into consecutive windows and develop a novel cross-window attention mechanism to ensure the consistency of global style and maximize the smoothness among windows. To achieve more accurate control, we extract the information from the source video via DDIM inversion and integrate the outcomes into the latent states of the generations. We also incorporate a video frame interpolation model to mitigate the frame-level flickering issue. Extensive empirical studies verify the superior efficacy of our method over competing baselines across scenarios, including the replacement of the attributes of foreground objects, style transfer, and background replacement. In particular, our method manages to edit videos with up to *128* frames according to user requirements.

## 1 Introduction

Diffusion models have gained tremendous popularity in image generation (Ho et al., 2022a; Ramesh et al., 2022; Nichol et al., 2021), with Stable Diffusion (Rombach et al., 2022) and Midjourney (Mid, 2023) as representative examples. They exhibit a remarkable ability to generate high-quality content and perform multimodal transformations. To further enhance the appeal and practicality of diffusion models, a natural idea is to extend them to create videos, which, yet, presents significant challenges, primarily arising from the increased complexity involved in modeling temporal information within videos and the need for a substantial number of annotated videos (Ho et al., 2022b; He et al., 2022).

Video editing is a simpler alternative to video generation and has seen notable advancements due to the introduction of pre-trained conditional diffusion models (Wu et al., 2022b; Qi et al., 2023; Ceylan et al., 2023; Khachatryan et al., 2023; Zhao et al., 2023). Despite the lower efficacy compared to methods trained directly on numerous videos, these methods can still yield realistic editing outcomes respecting user requirements. Their training-free nature is particularly desirable in vast user-specific scenarios with limited data.

The main challenge in training-free diffusion model-based video editing lies in effectively incorporating the modeling of temporal interplays between frames into plain diffusion models. Prior works address it by inflating 2D convolutions to pseudo 3D ones (Ho et al., 2022b) and extending the spatial self-attention to spatial-temporal one (Wu et al., 2022b; Qi et al., 2023). Yet, issues arise when confronted with long videos, where the coherence of the global style and the local subtleties cannot be reasonably maintained (Khachatryan et al., 2023; Qi et al., 2023; Zhao et al., 2023). Besides, there are instances where the intention is to edit a small object within the video but the existing methods also alter other factors (Khachatryan et al., 2023; Zhang et al., 2023; Zhao et al., 2023). These issues impede the practical application of existing training-free video editing methods.

This paper tackles these issues by proposing a simple and effective approach for text-driven training-free LOng Video Editing with ControlNet, named LOVECon. Technically, LOVECon follows the

basic video editing pipeline based on Stable diffusion and ControlNet (Zhang & Agrawala, 2023), with an additional step of splitting long videos into consecutive windows to accommodate limited computational memory. On top of these, we introduce a novel cross-window attention mechanism to maintain coherence in style and subtleties across windows. To ensure the structural fidelity to the original source video, we enrich the latent states of edited frames with information extracted from the source video through DDIM inversion (Song et al., 2020a). This guarantees that the content in the original video that we do not intend to edit remains unchanged. Additionally, LOVECon incorporates a video interpolation model (Huang et al., 2022), which polishes the latent states of the edited frames in the late stages of generation, to alleviate frame flickering. These techniques contribute to smoother transitions in long videos and significantly mitigate visual artifacts.

We gather 30 videos from Davis dataset (Pont-Tuset et al., 2017) and Pexels (pex, 2023) attached with source and editing prompts for empirical studies following (Qi et al., 2023; Zhao et al., 2023). We compare LOVECon against competitive ControlNet-based baselines (Khachatryan et al., 2023; Zhang et al., 2023; Zhao et al., 2023). We report metrics defined with CLIP (Radford et al., 2021) and from user studies. We observe LOVECon outperforms in both aspects of frame consistency and structural similarity with the original video. Besides, our method can edit long videos of up to 128 frames, delivering a smoother transition between frames compared to the baselines.

## 2 RELATED WORKS

### 2.1 DIFFUSION-BASED TEXT-TO-IMAGE GENERATION AND EDITING

Generative models like generative adversarial nets (GANs) (Goodfellow et al., 2014) and variational auto-encoders (VAEs) (Kingma & Welling, 2013) are typical approaches for image generation, but they suffer from unstable training or unrealistic generation. Recently, diffusion models (Ho et al., 2020; Nichol & Dhariwal, 2021; Song et al., 2020b) have emerged as a promising alternative to them, enjoying good training stability and showing outperforming sample quality. The application of diffusion models to text-to-image generation is particularly attractive due to its practical value (Nichol et al., 2021; Ding et al., 2022), with Imagen (Saharia et al., 2022), DALL-E2 (Ramesh et al., 2022), and Stable diffusion (Rombach et al., 2022) as examples.

Attempts have been made to leverage diffusion models for image editing based on user requirements. Prompt-to-Prompt (Hertz et al., 2022), Plug-and-Play (Tumanyan et al., 2023), and Pix2pix-Zero (Parmar et al., 2023) manipulate cross/self-attentions to achieve this, preserving the layout and structure of the source image. Blended Diffusion (Avrahami et al., 2022; 2023) proceeds by uniting the features of the source and generation images at each timestep. Null-text Inversion (Mokady et al., 2023) enables prompt-based editing by modifying the unconditional textual embedding for classifier-free guidance. ControlNet (Zhang & Agrawala, 2023) trains an extra encoder on top of the pre-trained diffusion model to incorporate additional information from the original image, such as canny edges or depth maps, as control information. However, directly applying the above method to videos without considering the temporal interplays can lead to inconsistent frames.

### 2.2 DIFFUSION-BASED TEXT-TO-VIDEO GENERATION AND EDITING

Due to limitations in computational resources and datasets, there are only a few large-scale text-to-video generative models (Ho et al., 2022b;a; Esser et al., 2023; Wu et al., 2022a) being developed. In contrast, diffusion-based video editing has gained increasing attention (Wu et al., 2022b; Qi et al., 2023; Zhao et al., 2023; Zhang et al., 2023; Yang et al., 2023; Hu & Xu, 2023; Chai et al., 2023), which benefits from the powerful pre-trained image generation models like Stable Diffusion. In particular, Wu et al. (2022b); Zhao et al. (2023) finetune the temporal-attention layers of the U-Net to maintain consistency in a short video clip. FateZero (Qi et al., 2023) follows Prompt-to-Prompt (Hertz et al., 2022) and fuses self-attention maps using blending masks obtained by DDIM inversion (Song et al., 2020a). Text2Video-Zero (Khachatryan et al., 2023) leverages cross-frame attention and enriches the latent states with motion dynamics to keep the global scene. Zhang et al. (2023) introduce an interleaved-frame smoother and a hierarchical sampler for long video generation. However, the generated videos of these existing methods often suffer from flickering issues. To address this, StableVideo (Chai et al., 2023) decomposes the input video into layered representations to edit separately and propagate the appearance information from one frame to the

next. Rerender-A-Video (Yang et al., 2023) performs optical flow estimation and only rerenders the keyframes for style transfer. CoDeF (Ouyang et al., 2023) uses a canonical content field and a temporal deformation field as a new type of video representation. Nonetheless, their pipelines involve extra model optimization. Unlike the existing approaches, our work keeps training-free, while addressing the frame-level flickering issues for long video editing.

## 3 PRELIMINARY

This section first reviews diffusion models (Ho et al., 2020) and latent diffusion models (Rombach et al., 2022). Then, we detail DDIM sampling and DDIM inversion (Song et al., 2020a), where the latter provides a typical approach for image editing.

**Diffusion models** gradually add Gaussian noises to some natural image $\boldsymbol{x}_0 \sim q(\boldsymbol{x})$ with the following transition probability

$$q(\boldsymbol{x}_t|\boldsymbol{x}_{t-1}) = \mathcal{N}(\boldsymbol{x}_t; \sqrt{\alpha_t}\boldsymbol{x}_{t-1}, \beta_t\mathbf{I}), t = 1, \ldots, T \tag{1}$$

where $\beta_t$ is the noise variance following an increasing schedule and $\alpha_t = 1 - \beta_t$. The forward process eventually makes $\boldsymbol{x}_T \sim \mathcal{N}(\mathbf{0}, \mathbf{I})$, i.e., the resulting image $\boldsymbol{x}_T$ becomes a white noise.

The generating process of diffusion models reverses the above procedure with the following $\theta$-parameterized Gaussian distribution:

$$p_\theta(\boldsymbol{x}_{t-1}|\boldsymbol{x}_t) = \mathcal{N}(\boldsymbol{x}_{t-1}; \mu_\theta(\boldsymbol{x}_t, t), \beta_t\mathbf{I}), t = T, \ldots, 1. \tag{2}$$

Ho et al. (2020) show that the mean prediction model $\mu_\theta(\boldsymbol{x}_t, t)$ can be parameterized as a noise prediction one $\epsilon_\theta(\boldsymbol{x}_t, t)$. $\epsilon_\theta$ is usually implemented as a U-Net due to its good image-to-image translation ability (Ronneberger et al., 2015). The training loss is the following mean-squared error:

$$\min_\theta \mathbb{E}_{t \sim \text{uniform}[1,T], \boldsymbol{x}_0 \sim p(\boldsymbol{x}), \boldsymbol{\epsilon} \sim \mathcal{N}(\mathbf{0},\mathbf{I})} ||\boldsymbol{\epsilon} - \epsilon_\theta(\sqrt{\bar{\alpha}_t}\boldsymbol{x}_0 + \sqrt{1 - \bar{\alpha}_t}\boldsymbol{\epsilon}, t)||^2, \text{ with } \bar{\alpha}_t := \prod_{i=1}^{t} \alpha_i. \tag{3}$$

**Latent Diffusion Models** shift the learning and generation from the image space to the latent space with the help of a discrete auto-encoder for high-resolution image generation. Specifically, the image $\boldsymbol{x}$ is first projected by the encoder to a low-dimensional latent representation $\boldsymbol{z} = \mathcal{E}(\boldsymbol{x})$. The denoising model operates in the latent space and is composed of self-attention layers as well as cross-attention layers to include textual guidance. The final latent states are mapped back to the image space by a decoder $\mathcal{D}$.

**DDIM Sampling** is a typical deterministic method for sampling from diffusion models. In the case of latent diffusion models, it follows

$$\boldsymbol{z}_{t-1} = \sqrt{\alpha_{t-1}}\frac{\boldsymbol{z}_t - \sqrt{1 - \alpha_t}\epsilon_\theta(\boldsymbol{z}_t, t, p)}{\sqrt{\alpha_t}} + \sqrt{1 - \alpha_{t-1}}\epsilon_\theta(\boldsymbol{z}_t, t, p), t = T, \ldots, 1, \tag{4}$$

where $\epsilon_\theta$ also accepts the text prompt $p$ for guiding the generation.

**DDIM Inversion** converts a clean latent $z_0$ back to the corresponding noise in reverse steps:

$$\boldsymbol{z}_{t+1} = \sqrt{\alpha_{t+1}}\frac{\boldsymbol{z}_t - \sqrt{1 - \alpha_t}\epsilon_\theta(\boldsymbol{z}_t, t, p)}{\sqrt{\alpha_t}} + \sqrt{1 - \alpha_{t+1}}\epsilon_\theta(\boldsymbol{z}_t, t, p), t = 0, \ldots, T - 1, \tag{5}$$

which contributes a reasonable starting point for image editing.

## 4 METHOD

As depicted in Figure 1, the proposed LOVECon leverages pre-trained Stable Diffusion (Rombach et al., 2022) to perform text-driven long video editing. LOVECon accepts the outcomes of DDIM inversion for the source frames as the starting point for sampling, and includes low-level details of the source frames as guidance with the help of ControlNet (Zhang & Agrawala, 2023). As the video may be long, the editing is performed window-by-window. Our technical contribution includes a cross-window attention mechanism, a latent information fusion strategy, and a frame interpolation

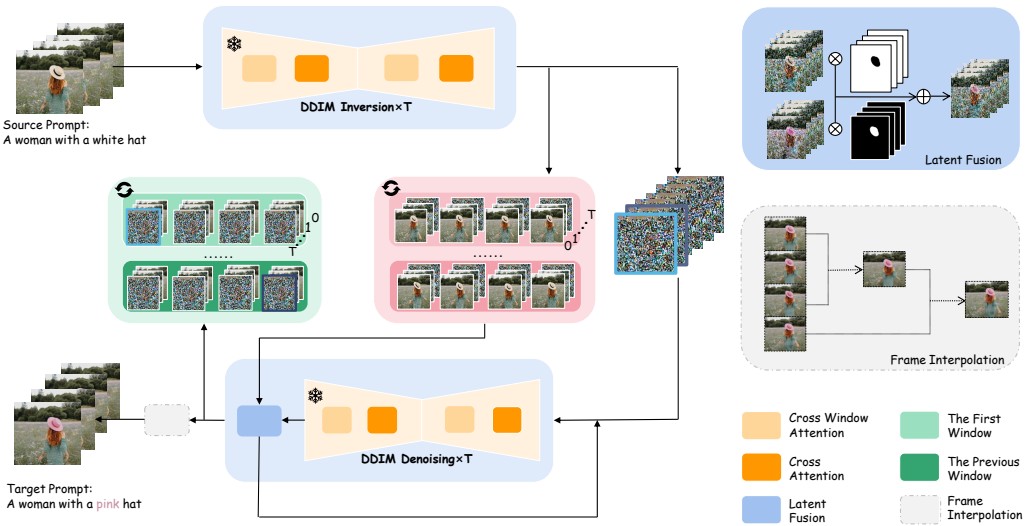

Figure 1: Method overview. LOVECon is built upon Stable diffusion and ControlNet (omitted in the plot for simplicity) for long video editing. LOVECon splits the source video into consecutive windows and edits sequentially, where cross-window attention is employed to improve inter-window consistency. LOVECon fuses the latent states of the edited frames with those of the source frames from DDIM Inversion to maintain the structure of the source video. LOVECon further incorporates a video interpolation model to address the frame-level flickering issue.

module. These components work collaboratively to overcome the shortcomings of existing methods in terms of generation length, temporal coherence, and fidelity to the source video. In the following, we provide detailed explanations for each component.

**Notations.** Given a video of $M$ frames, we evenly split it into multiple consecutive windows of size $K$, and use $\tilde{x}^{i,j}$ ($x^{i,j}$) to refer to the $j$-th source (edited) frame in the $i$-window, $i \in [1, M/K], j \in [1, K]$. The editing is governed by a source prompt $p_{src}$ and a target prompt $p_{tgt}$. $p_{obj}$ denotes the specific tokens in $p_{src}$ that indicate the editing objects. $\tilde{z}^{i,j}$ and $z^{i,j}$ denote the latent states of $\tilde{x}^{i,j}$ and $x^{i,j}$ in the latent diffusion models. $z_t^{i,j}$ denotes the latent state of the edited frame at $t$-th timestep and $\tilde{z}_t^{i,j}$ denotes that of the source frame from DDIM inversion. We consider using the noise prediction model stacked by attention layers (Vaswani et al., 2017), and we denote the features corresponding to $z_t^{i,j}$ in the model as $h_t^{i,j} \in \mathbb{R}^{L \times D}$ with $L$ referring to the sequence length.

## 4.1 CROSS-WINDOW ATTENTION

Long videos should be split into consecutive windows for editing due to resource constraints. However, naively treating the windows as isolated ones easily results in inconsistency between windows. The proposed cross-window attention addresses this issue by augmenting the generation of the current window with information from the other windows to ensure global style and to improve details.

Concretely, original self-attention applies computations to the $L$ tokens in $h_t^{i,j}$ to account for intra-image attention. Existing studies (Qi et al., 2023) refurbish it as a spatial-temporal one to include inter-image information for video editing, using $[h_t^{i,j}, h_t^{i,K/2}] \in \mathbb{R}^{2l \times d}$ to construct the key and value matrices for attention computation, where $h_t^{i,K/2}$ refers to the feature of the middle frame in the window of concern. However, such a strategy still cannot tackle the inter-window inconsistency.

We advocate further improving it by including more contextual information. Specifically, we extend $h^{i,j}$ to $[h_t^{1,1}, h_t^{i-1,K}, h_t^{i,j}, h_t^{i,K}] \in \mathbb{R}^{4l \times d}$ for computing the key and value matrices in the attention, where $h_t^{1,1}, h_t^{i-1,K}$, and $h_t^{i,K}$ denote the features of the first frame of the first window, the last frame of the previous window, and the last frame of the current window, respectively. The first one provides guidance on global style for the frames in the current window, and the others govern the variation dynamics. Compared to video editing methods relying on costly fully cross-frame attention (Zhang

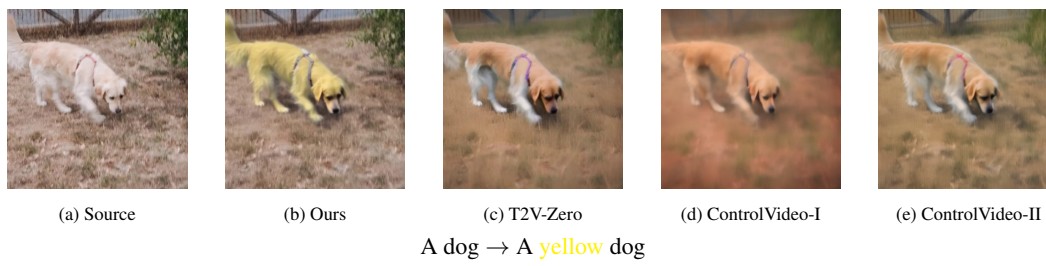

| (a) Source | (b) Ours | (c) T2V-Zero | (d) ControlVideo-I | (e) ControlVideo-II |

A dog → A yellow dog

Figure 2: Comparison on an image editing task. Baselines include T2V-Zero (Khachatryan et al., 2023), ControlVideo-I (Zhang et al., 2023), and ControlVideo-II (Zhao et al., 2023). The experiment setups are the same. As shown, our method has the capability to confine the semantic control of "yellow" specifically to the dog. In contrast, other methods suffer from semantic leakage.

et al., 2023), which performs attention over all frames in the window, our method can significantly reduce memory usage while keeping global consistency.

## 4.2 LATENT FUSION

To maintain the structural information of the source video, we propose to fuse the latent states of the source frames with those of the edited ones. Specifically, at $t$-th denoising step, there is:

$$z_t^{i,j} = m^{i,j} \odot z_t^{i,j} + (1 - m^{i,j}) \odot \tilde{z}_t^{i,j}, \tag{6}$$

where $\odot$ is the element-wise multiplication, $m^{i,j}$ denotes a time-independent mask to identify the regions that require editing, and $\tilde{z}_t^{i,j}$ denotes the outcomes of DDIM inversion for the source frames.

$m^{i,j}$ should not be specified manually as that can be laborious and time-consuming for long videos. Instead, inspired by prior studies (Qi et al., 2023; Hertz et al., 2022; Avrahami et al., 2022; 2023), we estimate $m^{i,j}$ by (1) collecting the cross-attention maps between textual features of $p_{obj}$ and visual features of $\tilde{z}_t^{i,j}$ during the DDIM inversion procedure, (2) binarizing the time-dependent cross-attention maps via thresholding and (3) aggregating them into a global binary one via pooling.

Nonetheless, Equation (6) totally discards the structural information of the source frames in the masked regions, but such information could be beneficial in the early stages of the reverse diffusion process. To mitigate this, we refine the latent fusion mechanism as:

$$z_t^{i,j} = m^{i,j} \odot [\gamma z_t^{i,j} + (1 - \gamma)\tilde{z}_t^{i,j}] + (1 - m^{i,j}) \odot \tilde{z}_t^{i,j}, \tag{7}$$

where $\gamma \in (0, 1)$ represents the trade-off coefficient and will be set to 1 after some timestep $T_0$.

In Figure 2, we showcase the efficacy of our latent fusion tactic in a ControlNet-based image editing task. As shown, our method enables accurate control and achieves high fidelity to the original image, while the baseline methods suffer from semantic leakage.

## 4.3 FRAME INTERPOLATION

While the cross-window attention and latent fusion mechanisms effectively preserve the global style and finer details, the generation may still suffer from frame-level flickering issues (Yang et al., 2023).

To address this, we introduce a video frame interpolation model to polish the latent states $z_t^{i,j}$. Given that interpolation models usually operate in the pixel space, we first project $z_t^{i,j}$ back to images via:

$$x_{t\to0}^{i,j} = \mathcal{D}(z_{t\to0}^{i,j}), \text{ with } z_{t\to0}^{i,j} = \frac{z_t^{i,j} - \sqrt{1 - \alpha_t}\epsilon_\theta(z_t^{i,j}, t, p_{tgt})}{\sqrt{\alpha_t}}. \tag{8}$$

Given these, the video interpolation model $\psi$ produces new frames via:

$$\dot{x}_{t\to0}^{i,j} = \psi(\dot{x}_{t\to0}^{i,j-1}, x_{t\to0}^{i,j+1}), j = 2, \dots, K - 1, \tag{9}$$

Table 1: Quantitative comparisons on the quality of editing outcomes. Refer to the main text for the meaning of the used metrics. In the user study, Text, Recon, FL, and Overall indicate text alignment, fidelity to the source video, temporal consistency, and overall impressions, respectively.

| Method | Objective Metrics | | | | User Study | | | |
|---|---|---|---|---|---|---|---|---|
| | CLIP-Text | CLIP-Temp | CLIP-SE | Con-L2↓ | Text | FL | Con | Overall |
| T2V-Zero | **0.2920** | 0.9830 | 0.7328 | 0.0035 | 0.08 | 0.08 | 0.05 | 0.06 |
| ControlVideo-I | 0.2860 | 0.9854 | 0.8353 | 0.0025 | 0.14 | 0.06 | 0.07 | 0.07 |
| ControlVideo-II | 0.2916 | 0.9826 | 0.8578 | 0.0038 | 0.12 | 0.08 | 0.06 | 0.07 |
| **Ours** | 0.2885 | **0.9889** | **0.8711** | **0.0023** | **0.66** | **0.78** | **0.82** | **0.81** |

where $\dot{\boldsymbol{x}}_{t\rightarrow0}^{i,1} := \boldsymbol{x}_{t\rightarrow0}^{i,1}$ and $\dot{\boldsymbol{x}}_{t\rightarrow0}^{i,K} := \boldsymbol{x}_{t\rightarrow0}^{i,K}$. $\dot{\boldsymbol{x}}_{t\rightarrow0}^{i,j}$ contains almost the same contents as $\boldsymbol{x}_{t\rightarrow0}^{i,j}$ while being more smoothing in the temporal axis (see Figure 5b). We map them back to the latent space via the encoder $\mathcal{E}$ of the latent diffusion model for the following sampling process.

Nonetheless, the repetitive use of the encoder and the decoder can lead to degradation in image quality and an accumulation of information loss (Yang et al., 2023). Therefore, we perform interpolation only two times, once at the last timestep of the reverse diffusion process within the window and once after the process within the whole video.

## 5 EXPERIMENT

### 5.1 SETTING

**Implementation Details.** We employ Stable Diffusion V1.5 (Rombach et al., 2022) and ControlNet V1.0 (Zhang & Agrawala, 2023) with canny edge maps, HED boundary, and depth maps as our base models. To perform frame interpolation, we utilize a trained video interpolation model named RIFE (Huang et al., 2022). For the dataset, we gather object-centric videos from DAVIS (Pont-Tuset et al., 2017) and Pexels (pex, 2023) following (Qi et al., 2023; Zhao et al., 2023). We design 30 cases for the experiment. The source videos are truncated to the first 48 frames, with $512 \times 512$ resolution. We manually annotate source and target prompts. The source prompts provide simple video descriptions, while the target prompts involve replacing or adding specific words to indicate the desired edits. We divide the editing task into three categories: the replacement of the attributes of foreground objects, style transfer, and background replacement. Style transfer is a global editing task, where we do not use masks for the latent fusion module. We empirically observe that performing latent fusion with Equation (7) in the late stages of the generation process can lead to poor object boundaries, so we invoke it only in the first $T_1$ timesteps, with details provided in Appendix D. We also undertake editing tasks on lengthy videos comprising up to 128 frames, which are also presented in detail in Appendix A.

**Baselines.** We take T2V-Zero (Khachatryan et al., 2023), ControlVideo-I (Zhang et al., 2023), and ControlVideo-II (Zhao et al., 2023) as baselines, which are all ControlNet-based video editing methods. We integrate DDIM inversion into T2V-Zero and ControlVideo-I to perform video editing rather than video generation. Additionally, we reimplement ControlVideo-II for long video editing. We employ a DDIM sampler with 50 steps, classifier-free guidance of magnitude 12, and a default control scale of 1 for all approaches for fair comparison.

**Metrics.** Following (Qi et al., 2023; Liu et al., 2023; Zhao et al., 2023), we compute **CLIP-Text**, which is the average of the CLIP feature similarities between the generated frames and the target prompt and indicates the video-text alignment. We also compute **CLIP-Temp**, which is the average of the CLIP feature similarities of consecutive frame pairs in the generation and indicates temporal consistency. We further propose **CLIP-SE** to assess the fidelity of the source video by computing the average of the CLIP feature similarities between all pairs of **s**ource and **e**dited frames. We empirically discover that CLIP-Temp cannot reliably identify the video flickering issue, so we introduce **Con-L2** as a supplementary metric, by calculating pixel-wise **L2** distance of all **con**secutive frame pairs in the edited video. In addition to objective metrics, we also conduct a user study to assess the alignment of the edited video quality with human preferences, covering text alignment, fidelity to the source video, temporal consistency, and overall impressions.

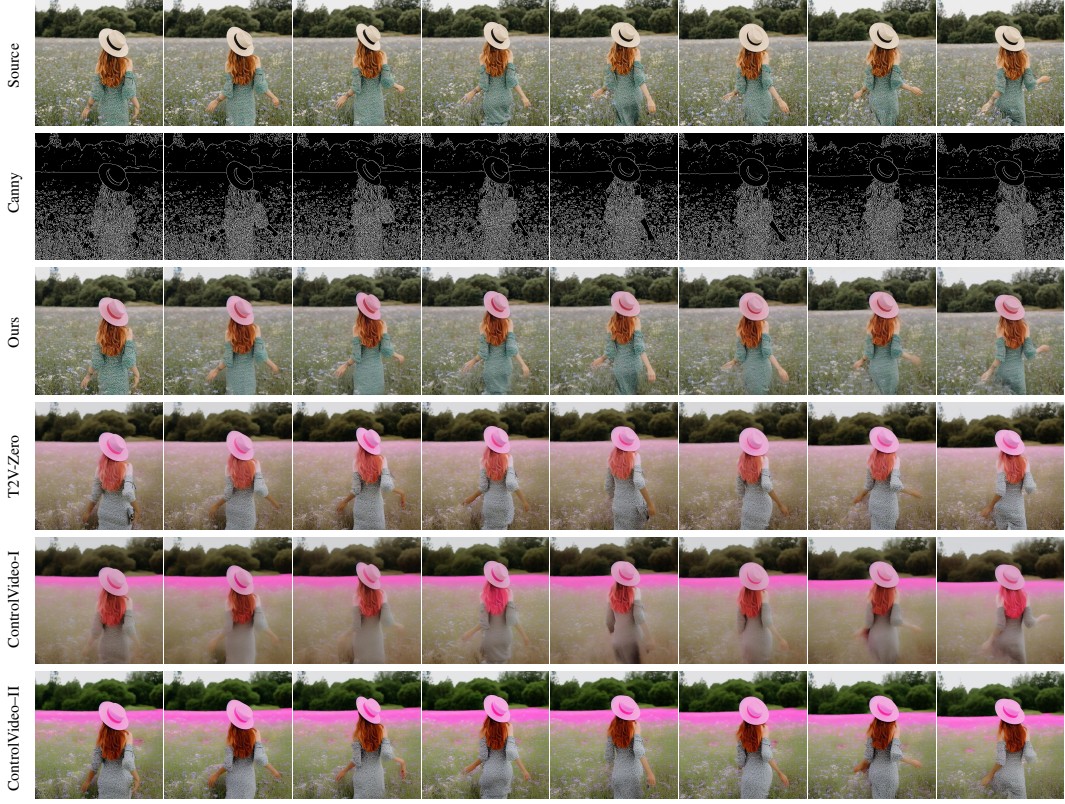

A woman with a white hat → A woman with a pink hat

Figure 3: Comparison between our method and baselines. Due to space constraints, we only select 8 frames evenly from the edited video of 48 frames. Refer to Appendix B for the complete edited video. The results reflect that LOVECon excels in providing precise control over the editing process, effectively recovering intricate details from the source frames, and presenting high fidelity.

## 5.2 QUANTITATIVE AND QUALITATIVE COMPARISONS

**Quantitative Comparisons.** Table 1 presents the quantitative comparisons between LOVECon and the baselines. As shown, LOVECon significantly outperforms the baselines in the aspect of CLIP-SE (i.e., fidelity), highlighting the effectiveness of latent fusion in our method. We also note that our approach slightly compromises the CLIP-Text metric, probably because our edited frames are more faithful to the source ones compared to the baselines. The superiority of our method in CLIP-temp and Con-L2 reflects that our edited video possesses smoother transitions and higher temporal consistency. Moreover, our method achieves significantly superior performance in the user study regarding all metrics, especially the overall impression, demonstrating that our generations are more preferred by the participants.

**Qualitative Comparisons.** Figure 3 illustrates the visual comparisons of the methods. While all of these methods can edit the video according to the target prompt, the visual quality varies. As shown, LOVECon has more precise control and can recover more details than the baselines, benefiting from the latent fusion strategy. T2V-Zero and ControlVideo-II can not preserve the details of hands. ControlVideo-I suffers from low image quality caused by blurred edges and unclear details. In addition to frame-wise comparison, our method maintains frame-level consistency in video format, while other methods exhibit significant flickering caused by subtle variations in minor details and colors (see Appendix B). In Figure 4, we present more qualitative results for a diverse range of editing scenarios. Through these examples, we demonstrate the exceptional ability of our method to preserve the desired editing effects while maintaining high fidelity and consistency.

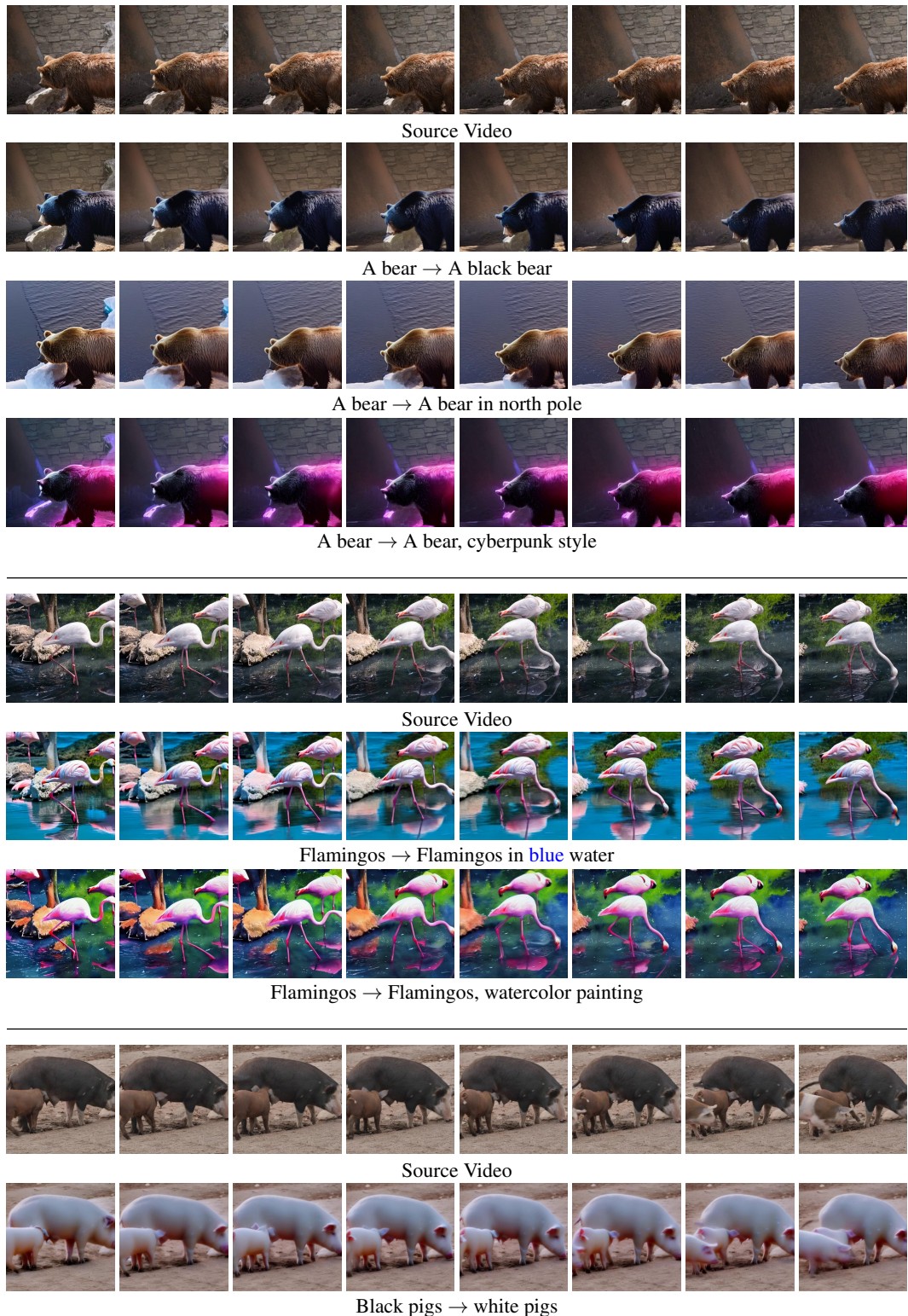

Source Video

A bear → A black bear

A bear → A bear in north pole

A bear → A bear, cyberpunk style

Source Video

Flamingos → Flamingos in blue water

Flamingos → Flamingos, watercolor painting

Source Video

Black pigs → white pigs

Figure 4: More editing results of our method using various prompts. These examples include attribute and background editing, and style transfer. The results demonstrate that our method maintains high fidelity and consistency when editing long videos while preserving the desired effects.

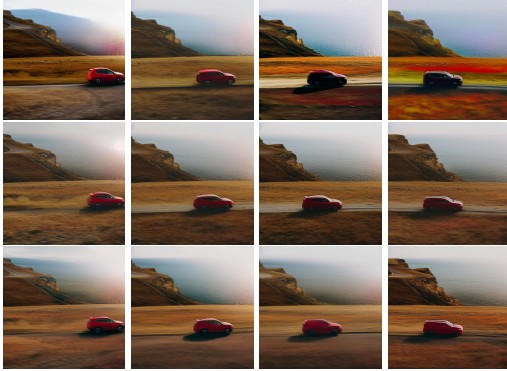 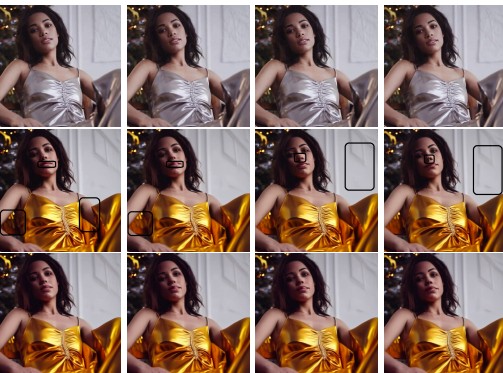

A car → A red car

A girl with a dress→A girl with a golden dress

(a) Ablation study on cross-window attention. The first row refers to editing the video window by window individually. The second and the third are edited with fully cross-frame attention-based ControlVideo-I using a hierarchical sampler and only with cross-window attention. We eliminate other modules for long video editing. These show that our pipeline with cross-window attention can achieve comparable results with the costly fully cross-frame attention.

(b) Ablation study on frame interpolation. The first row indicates the source frames, and the second and third are edited without and with the frame interpolation mechanism. Upon closer inspection of the consecutive frames, images in the second row exhibit subtle differences in color and details, annotated by the black boxes, which can lead to a decline in video quality. In contrast, those in the third row demonstrate more consistent texture and color.

Figure 5: Abation studies on cross-window attention and frame interpolation.

## 5.3 ABLATION STUDY

**Cross-window Attention.** Our method leverages cross-window attention to compensate for the absence of explicit temporal modeling and enhance global consistency. To assess its effectiveness, we perform an editing with the other introduced techniques eliminated. We exhibit the results in Figure 5a. Separate edits of each window based on naive cross-frame attention and the fully cross-frame attention-based ControlVideo-I (Zhang et al., 2023) using a hierarchical sampler are included as baselines. As shown, our proposed cross-window attention can improve the window consistency in the whole video and achieve comparable results with fully cross-frame attention while causing less computation overhead for attention.

**Frame Interpolation.** As stated, we leverage a frame interpolation model to improve frame-level consistency. To confirm its efficacy, we conduct an ablation study on it, where the latent fusion is excluded to enhance the clarity of the comparison. As Figure 5b shows, in the absence of the frame interpolation model, slight variations in the color and details (labeled in black boxes) among frames emerge, which causes flickering. This is particularly unacceptable in long videos. We notice such an issue is effectively ameliorated by the introduction of the frame interpolation model.

## 6 CONCLUSION

In this work, we propose a novel pre-trained diffusion models-based pipeline for long video editing that incorporates a cross-window attention mechanism to maintain temporal consistency. We introduce latent fusions for precise video editing and reevaluate the role of video interpolation models in ensuring video smoothness. All these techniques contribute to a significant improvement in video quality. Minimal computational resources are required during the editing process of long videos. We believe this opens up opportunities for more ordinary individuals to engage in long video editing and explore further possibilities of conditional diffusion models.

**Limitations & Future Work.** While our method gets smooth and consistent edited long videos, it still has limitations. One limitation is that ControlNet restricts shape changes, allowing only modifications with similar shapes. Another limitation is that we have observed suboptimal editing results when there are significant content changes in the source video, e.g., movements. In the future, we will focus on further improvements to enhance the visual experience of the edited video.

ETHICS STATEMENT

This work proposes a training-free ControlNet-based video editing method. The potential negative impact could involve individuals exploiting it for harmful purposes, such as violating personal rights, infringing upon image rights, or generating unsettling content.

REPRODUCIBILITY STATEMENT

We submit our codes in the supplementary material, and details of hyperparameters are presented in the Appendix section.

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

# A    QUALITATIVE RESULTS ON LONG VIDEO OF 128 FRAMES

We present a long video editing example of 128 frames. The first row is the source frames, followed by the edited frames. The results demonstrate that our method is capable of handling the task of long video editing and preserving long-time temporal consistency.

A car → A red car

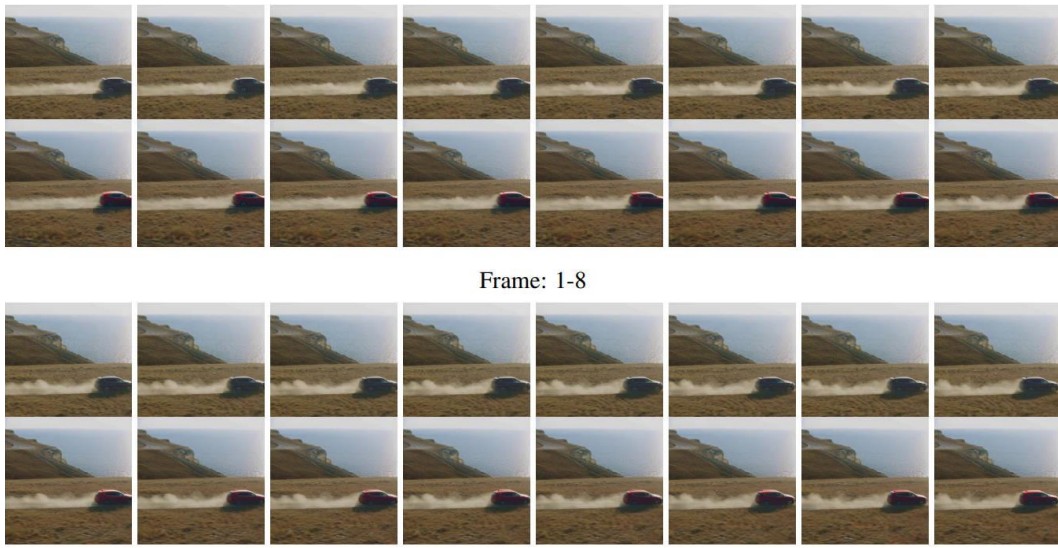

Frame: 1-8

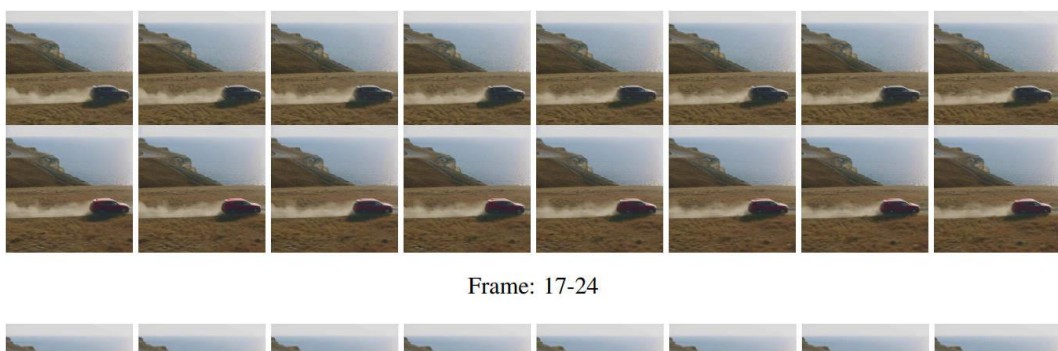

Frame: 9-16

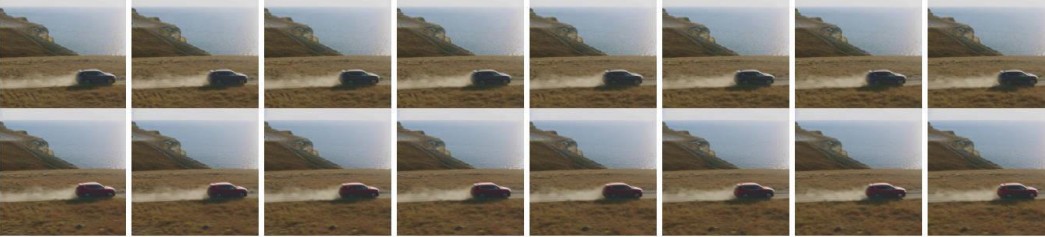

Frame: 17-24

Frame: 25-32

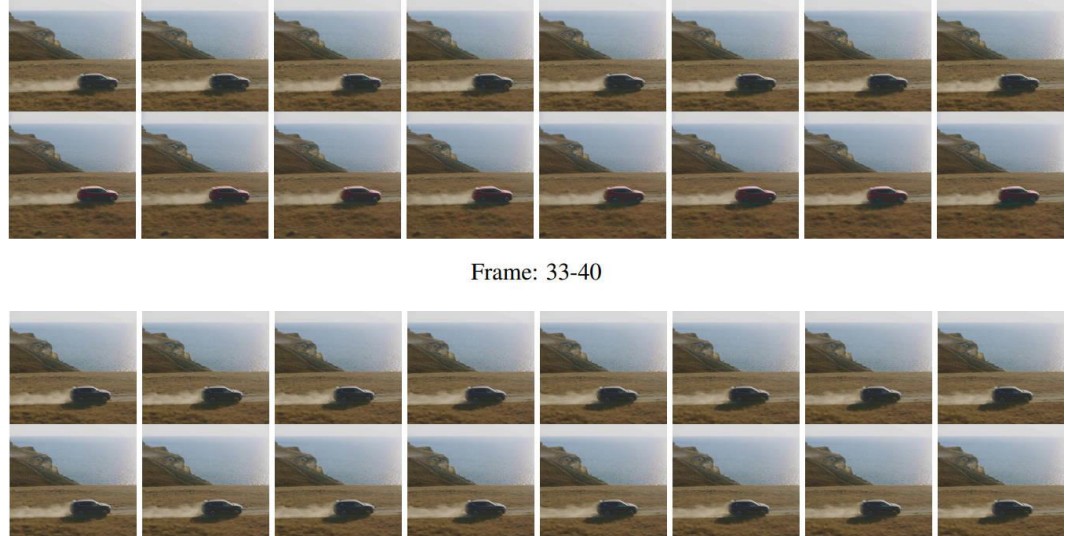

Frame: 33-40

Frame: 41-48

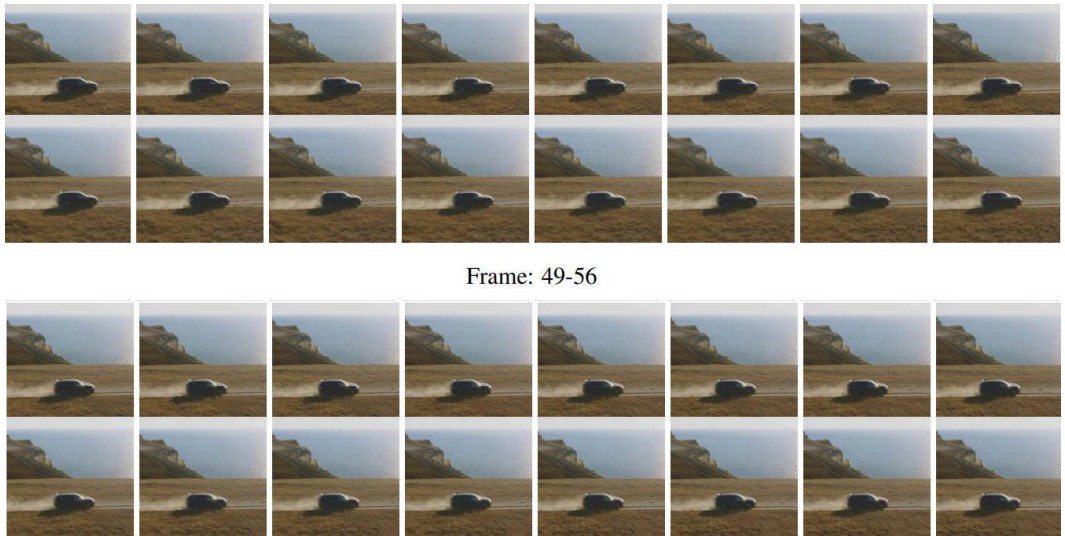

Frame: 49-56

Frame: 57-64

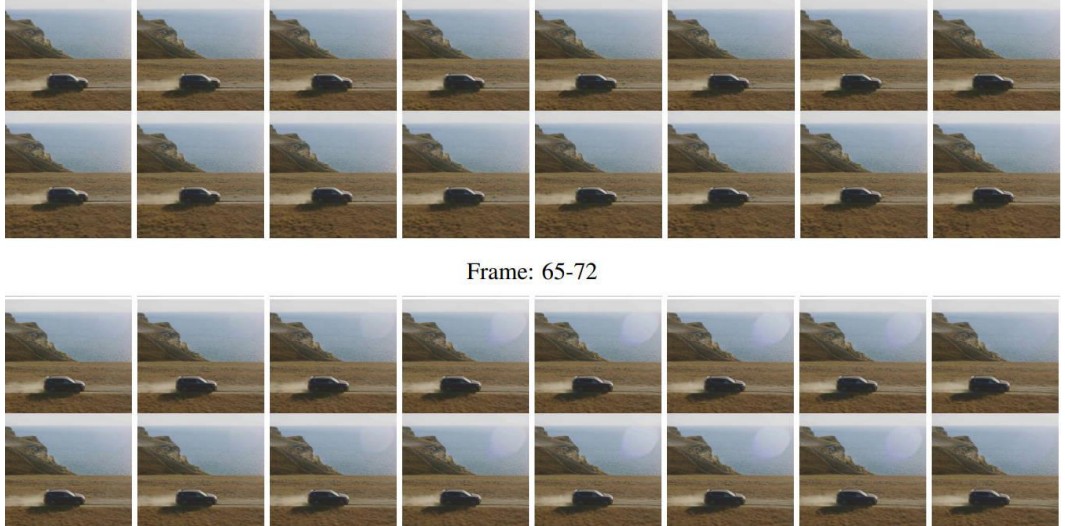

Frame: 65-72

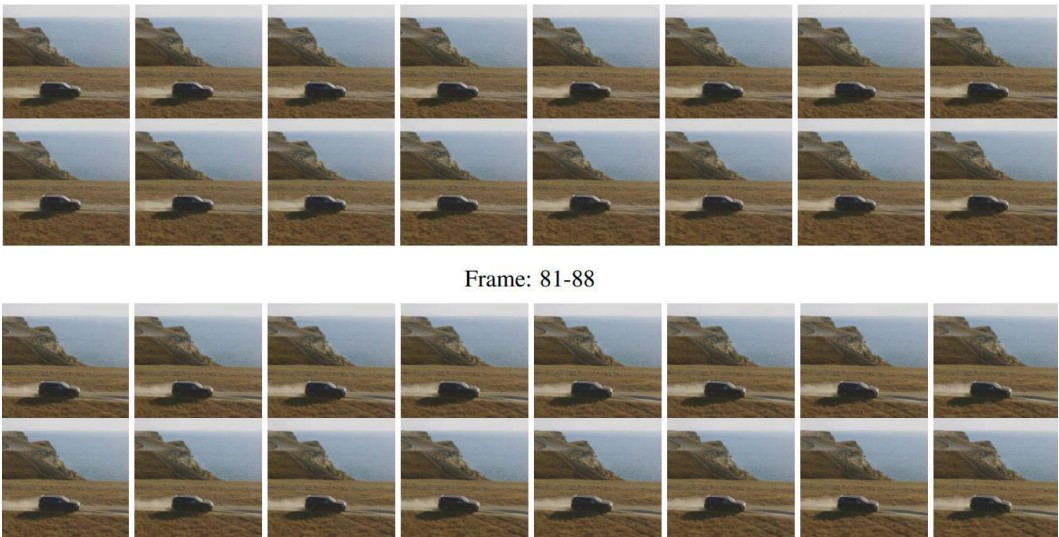

Frame: 73-80

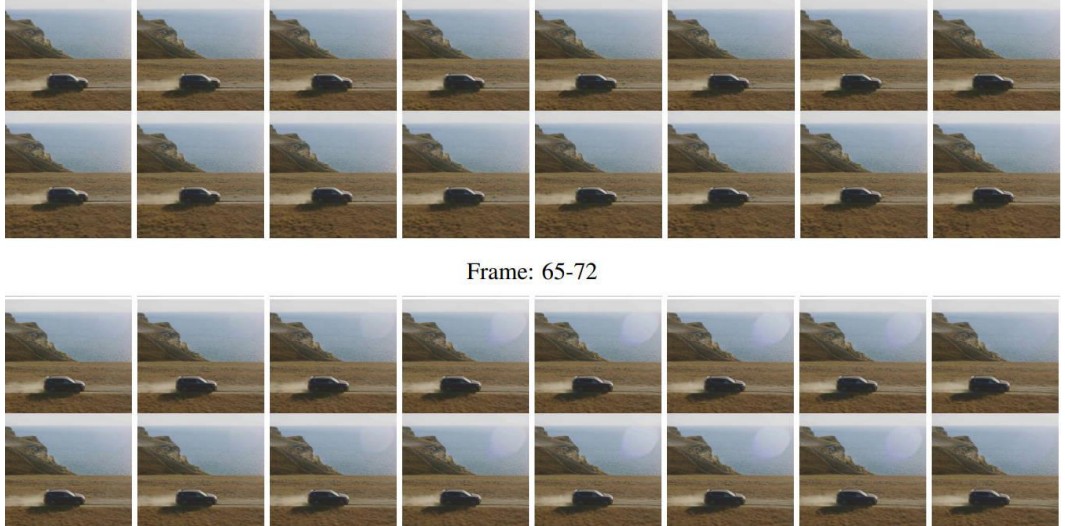

Frame: 81-88

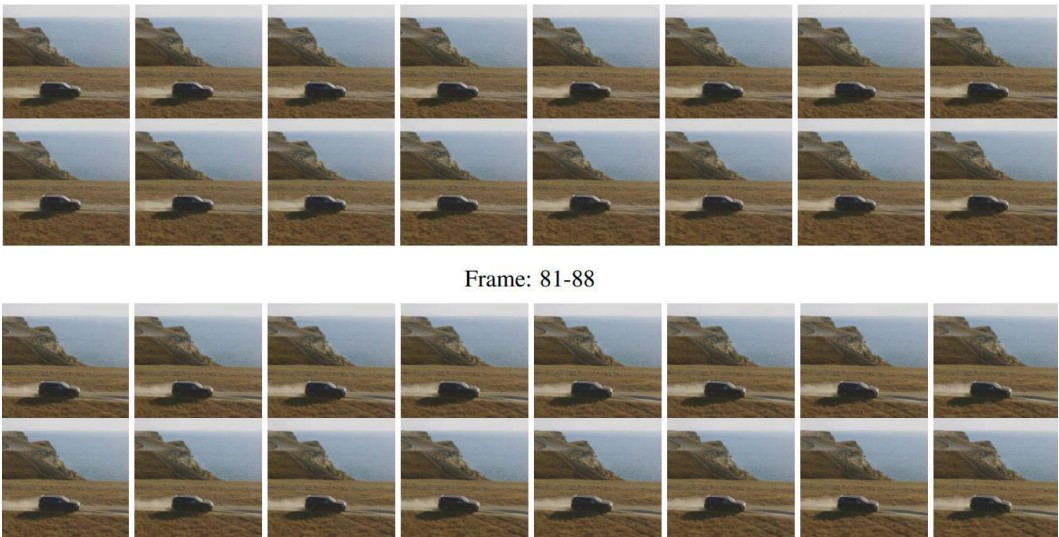

Frame: 89-96

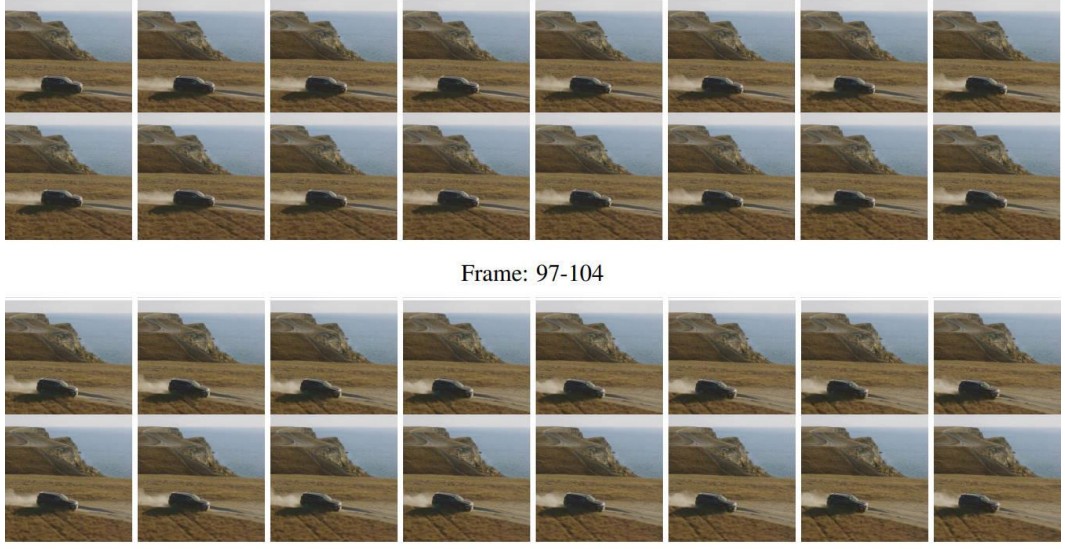

Frame: 97-104

Frame: 105-112

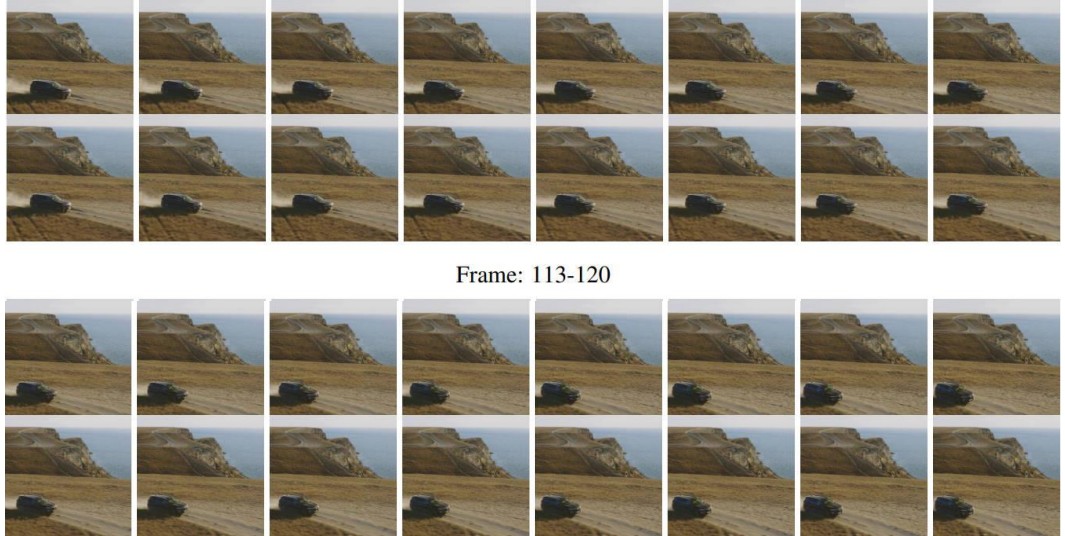

Frame: 113-120

Frame: 121-128

## B    COMPARISON FOR OUR METHOD AND BASELINES

Here we provide the complete comparison results.

A woman with a white hat → A woman with a pink hat

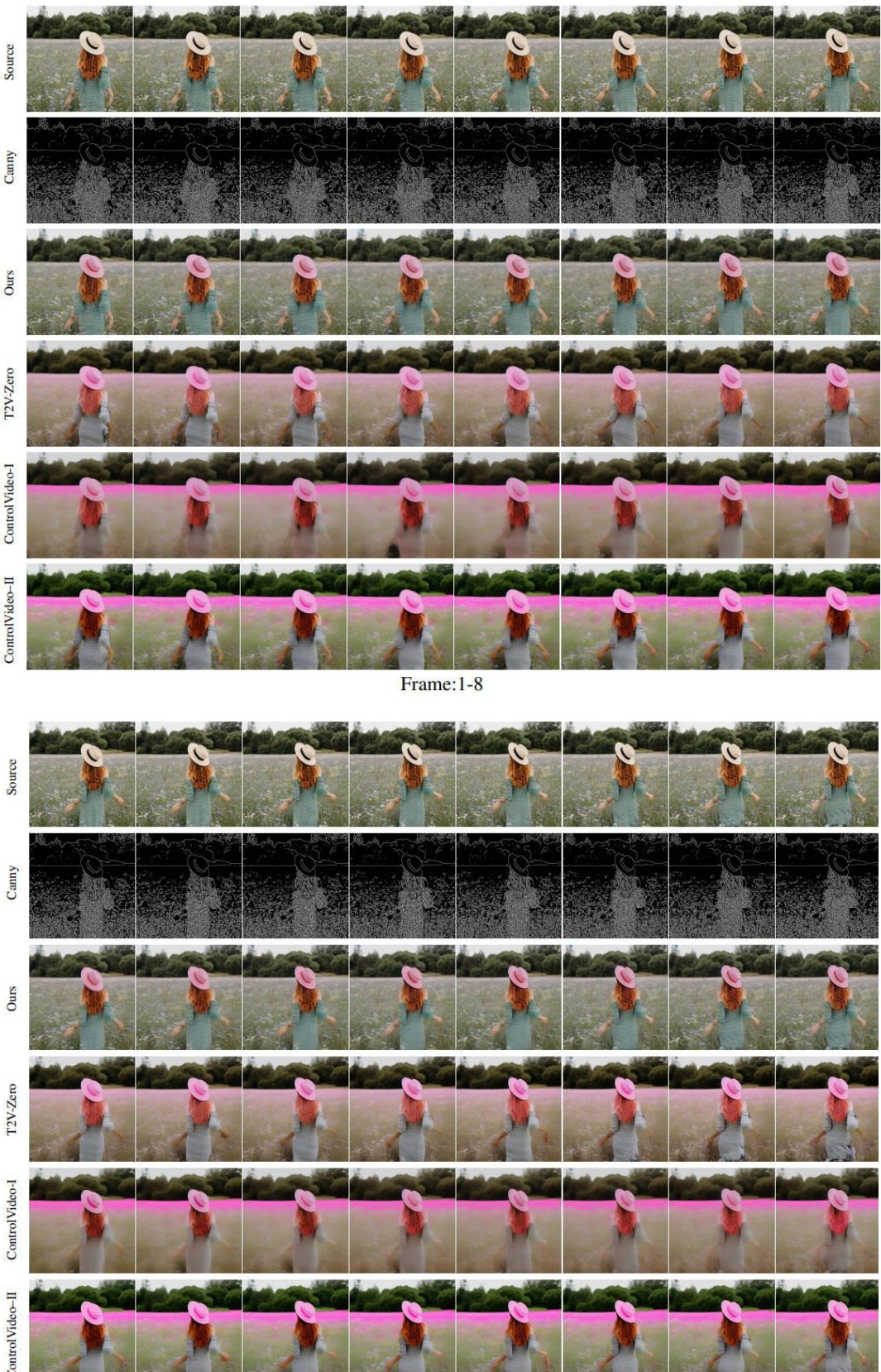

Frame:1-8

Frame:9-16

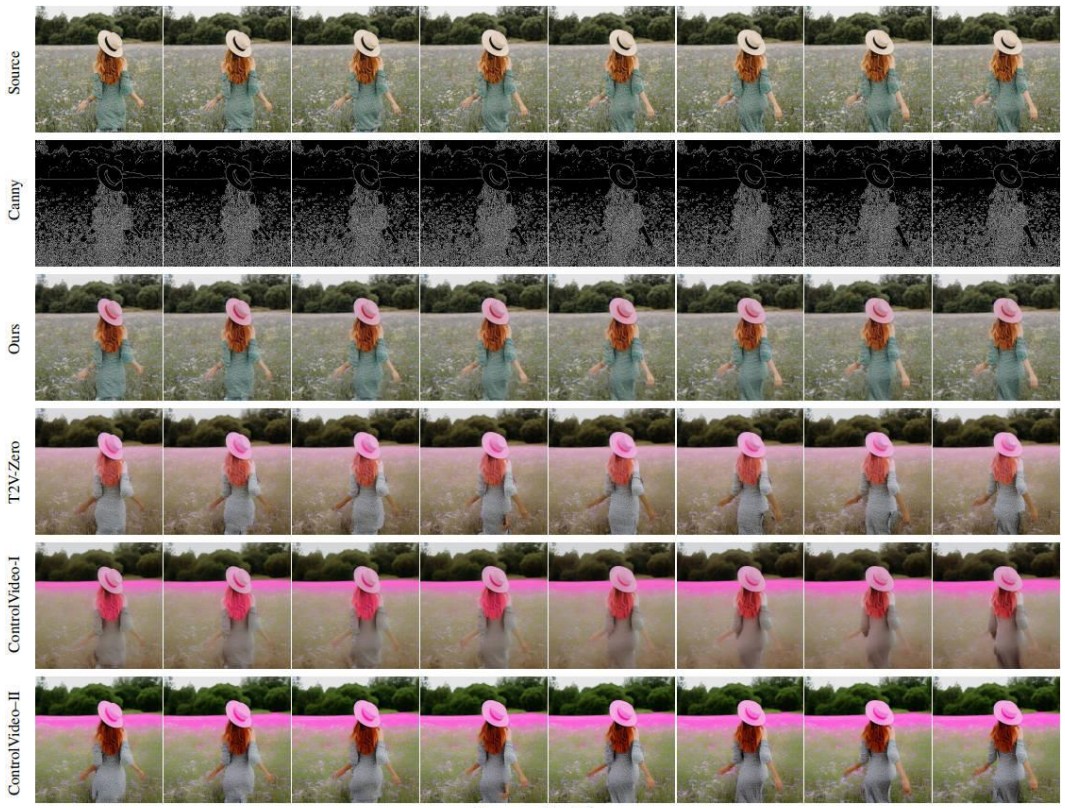

Frame:17-24

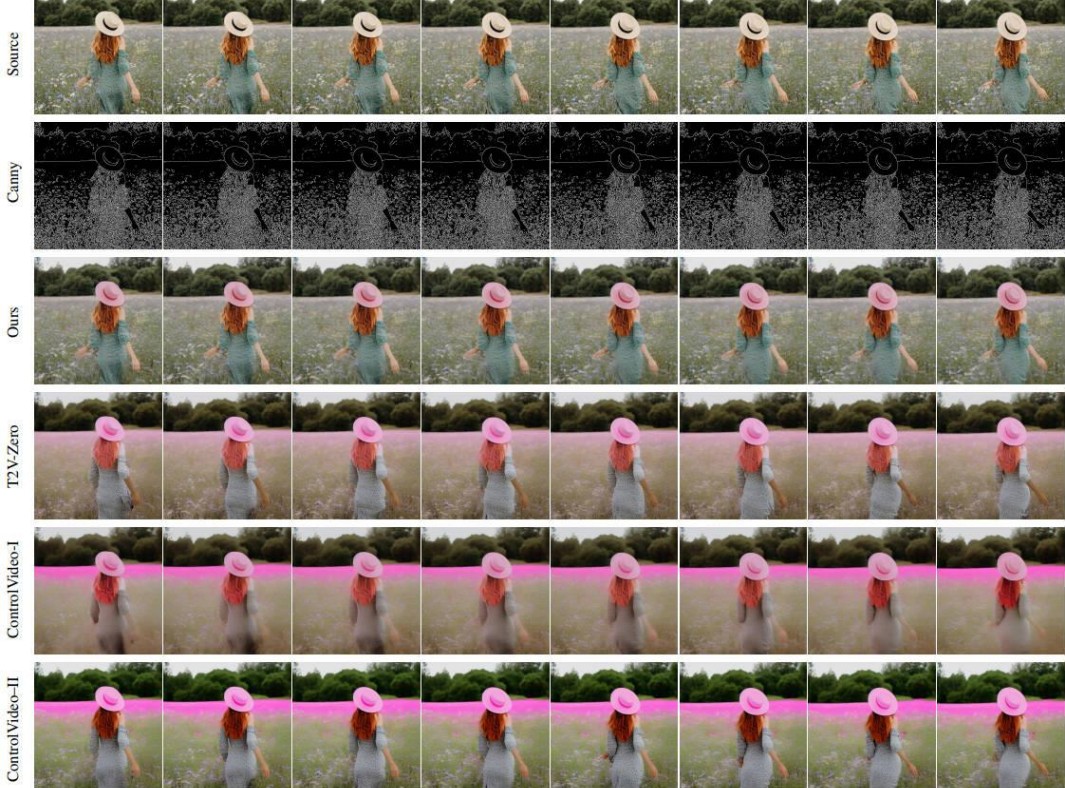

Frame:25-32

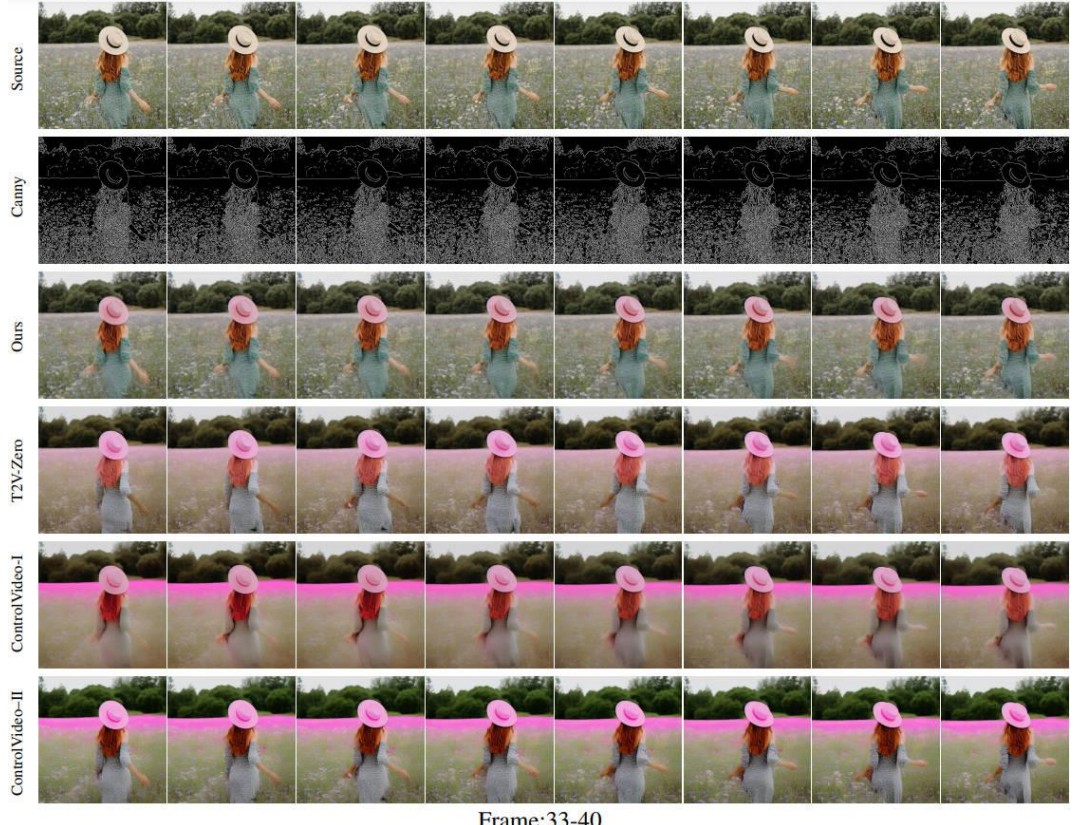

Frame:33-40

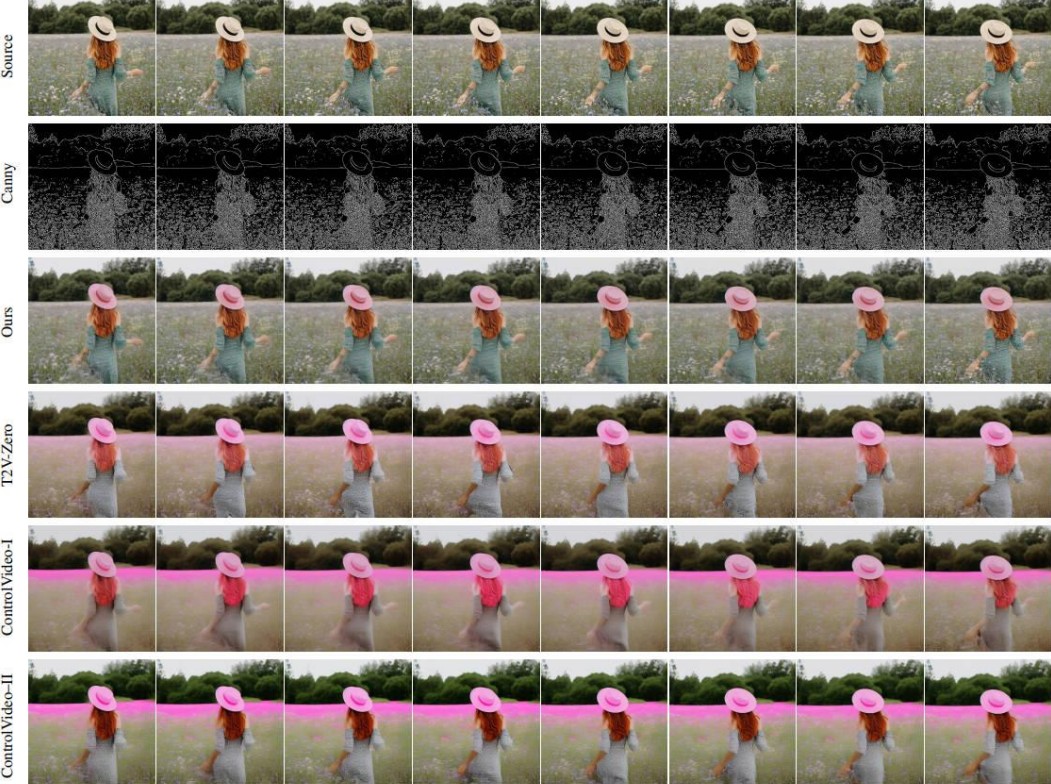

Frame:41-48

## C    USER STUDY

In the user study, a total of 15 participants took part in the survey. We evaluated the quality of the videos based on four dimensions: text alignment, fidelity, temporal consistency, and overall impression. The participants first watched the original video and became acquainted with the editing target, and then selected the best video for each criterion in each case. After the survey concluded, we computed the ratio of each criterion to the total count for comparison purposes.

## D    HYPERPARAMETER

In latent fusion, we can design different fusion strategies for different editing tasks.

- Replacement of the attributes of the foreground objects: we fuse masks for the first 40 steps for preserving the background and fuse latent states of source frames for the first 30 steps for maintaining the details of the foreground object empirically, i.e., $T_0 = 30, T_1 = 40$, and we set $\gamma = 0.97$.

- Style transfer: we do not fuse masks in the task and fuse latent states of source frames for the first 10 steps for providing rough source information, i.e., $T_0 = 0, T_1 = 10$, and we also set $\gamma = 0.97$.

- Background replacement: we fuse masks for preserving the foreground objects and fuse latent states of source frames for the first 10 steps for providing rough source information, i.e., $T_0 = 0, T_1 = 20$, and $\gamma = 0.97$.

These are the default settings. For different source videos, editing prompts and tasks, we can get a better edited video by tuning hyperparameters.

