# OpenReview forum: "LOVECon: Text-driven Training-free Long Video Editing with ControlNet"
_ICLR.cc/2024/Conference — Submitted to ICLR 2024_

### Official Review · Reviewer_WhZ8 · 2023-10-28

**Soundness:** 3 good
**Presentation:** 3 good
**Contribution:** 2 fair
**Rating:** 5
**Confidence:** 5

**Summary:**

This paper focuses on long video editing with training-free diffusion models.  The authors propose a useful cross-window attention mechanism to ensure the consistency and length of the video. They also leverage DDIM for accurate control and a video frame interpolation model to mitigate the frame-level flickering issue. The authors presented rich and excellent experimental results, and provided their code and products in supplementary materials.

**Strengths:**

Long video editing, consistency maintenance, and structural fidelity are fundamental issues in text guided long video editing. The authors propose a simple and systematic solution for training free long video editing. The authors present a good number of experiments validating the effectiveness of their approach. The paper is overall well written and easy to follow.

**Weaknesses:**

1. The authors have pieced together too many other people's methods to achieve the goals, so their model capabilities are limited, such as being unable to perform shape transformations, object additions or deletions. So their products are limited to color or style changes.

2. The innovation of the methods proposed by the authors is limited, as they have not effectively established the temporal information of the video. The so-called cross-window is naive (a minor change of ControlVideo **[1]**) and may be helpful for local video smoothing, but it still cannot truly establish the temporal dependence of long videos. This is reflected in the experimental results that although the generated video actions are locally coherent, the overall appearance is somewhat strange.

**[1]** Zhang, Yabo, Yuxiang Wei, Dongsheng Jiang, Xiaopeng Zhang, Wangmeng Zuo and Qi Tian. “ControlVideo: Training-free Controllable Text-to-Video Generation.” ArXiv abs/2305.13077 (2023): n. pag.

**Questions:**

Cross-window attention is proposed to improve inter-window consistency, then how do the authors ensure consistency within the window?
As shown in Figure 5 (a), fully crossframe attention-based ControlVideo not only maintains the loyalty of the background, but also maintains the temporal consistency of the target subject. On the contrary, the author's method blurs the background and changes the target subject in the temporal sequence (the car window turns red).

**Details Of Ethics Concerns:**

No ethics concerns.

---

> ### Author Response · Authors · 2023-11-16
> **Thank you for the valuable feedback!**
>
> We would like to express our gratitude to Reviewer **WhZ8** for reviewing our paper and raising valuable questions. Below we respond to the questions. We would highly appreciate it if the reviewer agree with our response and consider raising the score. Thank you so much!
>
> **Q1. The model capabilities are limited, such as being unable to perform shape transformations, object additions, or deletions.**
>
> **Please first refer to the general reply for the clarification of our contributions.** We also clarify that the inability to perform shape transformations, object additions, or deletions does not stem from the fact that our model capacities are not enough. The related works [1, 2, 3] can neither achieve that because we all hinge on pre-trained diffusion models and controlnets for **training-free video editing**. Besides, as argued, one of our main goals is to keep the fidelity of the generations to the source videos, instead of creating new content. We sincerely hope the reviewer can re-evaluate the contribution of this work based on these clarifications.
>
> **Q2. The innovation is limited, and cross-window cannot establish the temporal dependence of long videos？**
>
> Thanks for sharing your concerns. We would like to clarify again that we extend the sparse attention mechanism to cross-window attention for long video editing, while vanilla sparse attention or cross-frame attention leads to noticeable transitions between windows. Fig 5 a) shows the result of the separate edit of each window based on naive cross-frame attention. From the experiments, **we can conclude that the cross-window can establish the temporal dependence of long videos to some extent and alleviate the different styles between windows.**
>
> **Q3. Consistency within the window, and worse performance than fully cross-frame attention-based ControlVideo?**
>
> Thanks for digging into the details. In cross-window attention, we **incorporate the features of the last frame in the current window in the KV matrix**, which can ensure consistency within the window. In fact, the hierarchical sampler in ControlVideo serves the same purpose of establishing the temporal dependence of long videos. Our cross-frame attention is a replacement for it in an auto-regressive pipeline. We also clarify that **it is natural to get better results when incorporating more features (as done by fully cross-frame attention-based ControlVideo), but we argue that we can get comparable results with less computation overhead and quite better results than vanilla sparse attention, and visualization results are not cherry-picked.** Besides, with our complete pipeline, we perform better than baselines, which shows that each part makes a difference to the final effects, especially the latent fusion module can provide more accurate controls and further improve consistency (see car_comparision.mp4 in the updated supplementary for the comparison results of the "a red car" case).
>
> [1] Zhao et.al. ControlVideo: Adding Conditional Control for One Shot Text-to-Video Editing
>
> [2] Khachatryan et.al. Text2Video-Zero: Text-to-Image Diffusion Models are Zero-Shot Video Generators
>
> [3] Zhang et.al. ControlVideo: Training-free Controllable Text-to-Video Generation

---

> > ### Author Response · Authors · 2023-11-21
> > **Sincerely looking forward to the further discussions**
> >
> > Dear reviewer,
> >
> > We are wondering if our clarification of contributions and response have effectively resolved your concerns. If our response has addressed your concerns, we sincerely hope for your re-evaluation of our contributions and consideration in raising your score.
> >
> > If you have any additional questions or suggestions, we would be happy to have further discussions.
> >
> > Best regards,
> >
> > The Authors

---

### Official Review · Reviewer_59Nz · 2023-10-29

**Soundness:** 2 fair
**Presentation:** 3 good
**Contribution:** 2 fair
**Rating:** 5
**Confidence:** 4

**Summary:**

This paper aims to address the problems of limited video length and temporal inconsistency in text-driven video editing task.
It introduces a training-free pipeline based on ControlNet (i.e., LOVECon) for efficient long video editing, including three key designs.
In specific, LOVECon develops a novel cross-window attention mechanism to ensure the consistency of global style.
Secondly, it fuses the latents of source video to obtain more accurate control.
Finally, it incorporate a video frame interpolation model to deflicker the generated videos.

**Strengths:**

See summary.

**Weaknesses:**

1. The main contribution of long-video editing is limited in paper. The cross-window attention mechanism is common in long video editing/generation [1].
2. Question about visualization results of ControlVideo-II [2] in Fig.3. ControlVideo-II manages to achieve fine-grained control in original paper[2] (e.g., hair color), but fails to do so in Fig.3. Could the authors explain the reason?

[1] Fu-Yun Wang, Wenshuo Chen, et al. Gen-L-Video: Multi-Text to Long Video Generation via Temporal Co-Denoising.
[2] Min Zhao, Rongzhen Wang, et al. Controlvideo: Adding conditional control for one shot text-to-video editing.

**Questions:**

See weaknesses.

---

> ### Author Response · Authors · 2023-11-16
> **Thank you for the valuable feedback!**
>
> We would like to extend our appreciation to Reviewer **59Nz** for taking the time to review our paper and posing questions. Below we respond to the questions. We would highly appreciate it if the reviewer agree with our response and consider raising the score. Thank you so much!
>
> **Q1. The cross-window attention mechanism is common in long video editing/generation in Gen-L-Video.**
>
> Thanks for pointing out the connection between our method and Gen-L-Video. From our limited understanding, Gen-L-Video proposes bidirectional cross-frame attention, which is used with a short video clip instead of the whole video, and suggests learning clip identifier $e^i$ for each short video clip to guide the model in denoising the corresponding clip.  But **our proposed cross-window attention is for the whole video**, which is the main difference.
>
> In this field, cross-frame attention is widely used and different papers set different keyframes but we think they serve the same purpose. However, for the whole video consistency when dealing with long videos, we design the cross-window attention mechanism. **Refer to the general reply for more clarifications of our contributions.**
>
> **Q2. ControlVideo-II manages to achieve fine-grained control in the original paper (e.g., hair color), but fails to do so in Fig.3.**
>
> Thanks for noticing the minor differences. In the original codebase of ControlVideo-II, it can only generate 8 frames at a time. For a fair comparison, when generating longer videos, we concatenated the first frame as references (i.e., including the cross-window attention to some extent), and we didn't change other parts. **The visualization results are from the codes with modifications mentioned above.**  If you would like to check the authenticity, you may check the ControlVideo-II codes and generate the video for comparison. The source video is in the supplementary.

---

> > ### Author Response · Authors · 2023-11-21
> > **Sincerely looking forward to the further discussions**
> >
> > Dear reviewer,
> >
> > We are wondering if our clarification of contributions and response have resolved your concerns. If our response has addressed your concerns, we hope for your reconsideration in raising your score.
> >
> > If you have any additional questions or suggestions, we would be happy to have further discussions.
> >
> > Best regards,
> >
> > The Authors

---

### Official Review · Reviewer_EDK1 · 2023-11-01

**Soundness:** 2 fair
**Presentation:** 3 good
**Contribution:** 2 fair
**Rating:** 5
**Confidence:** 4

**Summary:**

This paper proposed a new long video editing method based on video controlnet and text-to-image latent diffusion models. The main contributions of the proposed method are 1) a cross-clip sparse attention to ensure the consistency of the long video while saving memory, 2) a latent fusion mechanism based on attention maps of the editing target, and 3) a frame smoothing mechanism based on near frame interpolation. The experimental results on 30 videos and the CLIP-based metrics demonstrate that the proposed method has better temporal consistency and video quality than the compared baselines.

**Strengths:**

- [writing] This paper is easy to follow, and the overall writing is good to me.
- [method] This work tackles the challenging task that editing long videos.
- [experiment] The visual results show that the proposed method allows the editing on the target area while maintaining the other image regions intact. Quantitative results also verify that the proposed method has better video quality and temporal consistency.

**Weaknesses:**

- [method] The first drawback of the proposed method is that it is very similar to the VideoControlNet, where the main difference is the modification to tackle long video editing cases. 1) The cross-window attention is a special sparse attention mechanism, which has been widely used by other vision models. 2) The latent fusion module has also been used in other works. 3) The frame interpolation mechanism is a long-history video frame smoothing approach. Although combining existing approaches to solve a challenging problem is meaningful, I am hesitant to give a high score on this work since the "added" modules are all very common and bring limited insight into the community.
- [experiment] In the visual comparison, I find the main visual benefit of the proposed method is that it only edits the target region while keeping the other part intact, which, according to my understanding, is contributed by the latent fusion module, making it hard to evaluate the effectiveness of the cross-window attention. I think more visual (like fig. 5a) and quantitative comparisons without the latent fusion would help.
- [experiment] The experiments are only based on 30 videos, which is far from enough to show the robustness of the proposed method. Moreover, I feel CLIP-based metrics may not be reliable enough to evaluate the overall quality of the generated videos. IS and FVD (if having a larger evaluation set as the reference) should be used for the quality comparison.

**Questions:**

No

---

> ### Author Response · Authors · 2023-11-16
> **Thank you for the valuable feedback!**
>
> We thank Reviewer **EDK1** for offering suggestions for improvements. Below we respond to the questions. We would highly appreciate it if the reviewer agree with our response and consider raising the score. Thank you so much!
>
> **Q1. The method is similar to the VideoControlNet, and modules are very common?**
>
> Thanks for pointing out such a connection. Here we clarify that in VideoControlNet, the editing process for a video involves multiple stages. They acquire inpainting masks based on optical flows, generate the keyframes, and interpolate the inner frames irrelevant to the diffusion pipeline. This resembles the hierarchical sampler in [1]. Instead, we edit the whole video with our diffusion pipeline auto-regressively. We think **the two methods are fundamentally different. Please refer to the general response for more clarifications of our contributions.**
>
> **Q2. Influenced by the latent fusion module, it is hard to evaluate the effectiveness of the cross-window attention?**
>
> Thanks for the suggestion. We **add a new baseline** in Fig 5 a), the separate edit of each window based on naive cross-frame attention. **Note that the experiments do not include a latent fusion module.** Comparing our cross-window attention without the latent fusion module to this baseline, we can **conclude that cross-window attention can effectively improve consistency in long video editing.**
>
> **Q3. Experiments don't show the robustness and CLIP-based metrics are not reliable?**
>
> Thanks for pointing it out. We clarify that the number 30 was carefully chosen after thoughtful consideration. In the quantitative comparison in [2], only 25 videos were selected for comparisons with baselines. In [3], 40 videos from the Davis dataset and other videos from the wild were used for comparisons.
>
> Regarding metrics, unfortunately, there is currently no standardized validation set or evaluation metric available. Therefore, we have followed the practice of utilizing metrics from the CLIP series and conducting user experiments, which align with previous works like [4] in this field. We also add Con-L2 as a metric to evaluate consistency. We welcome more specific suggestions on the metric.
>
> [1] Yang et.al. Rerender A Video: Zero-Shot Text-Guided Video-to-Video Translation
>
> [2] Khachatryan et.al. Text2Video-Zero: Text-to-Image Diffusion Models are Zero-Shot Video Generators
>
> [3] Zhao et.al. ControlVideo: Adding Conditional Control for One Shot Text-to-Video Editing
>
> [4] Qi et.al. FateZero: Fusing Attentions for Zero-shot Text-based Video Editing

---

> > ### Author Response · Authors · 2023-11-21
> > **Sincerely looking forward to the further discussions**
> >
> > Dear reviewer,
> >
> > We are wondering if our clarification of contributions and response have resolved your concerns. If our response has addressed your concerns, we would highly appreciate it if you could re-evaluate our work and consider raising the score.
> >
> > If you have any additional questions or suggestions, we would be happy to have further discussions.
> >
> > Best regards,
> >
> > The Authors

---

### Author Response · Authors · 2023-11-16
**General reply**

We thank all the reviewers for the constructive feedback and valuable suggestions. Here we make some clarifications of our contributions and add a new ablation study regarding cross-frame attention. We also add some comparison videos with baselines in the updated supplementary to show the improvements in the video editing. We genuinely hope all the reviewers can re-evaluate the contributions of this work based on these clarifications and experiments.

**Clarification of contributions**:
1) **Cross-window Attention.** We extend the sparse attention mechanism to cross-window attention for long video editing. As we split frames into windows due to memory constraints, our cross-window attention incorporates features from the previous windows to improve temporal consistency of the generation, while **vanilla sparse attention or cross-frame attention easily leads to abrupt transitions between windows (as shown in our ablation)**.
2) **Latent Fusion.** Though the concept of latent fusion has been investigated previously, we find critical empirical tricks (such as the scaling mechanism in Eq 7) to make it applicable to diffusion models for video editing.
3) **Frame Interpolation**. Empirically, for editing videos with only 8 frames, on which previous works focus, the pixel-level flickering may not be noticeable. However, as the duration of the video increases, such an issue can significantly impact the viewing experience. To maintain the training-free nature of the pipeline, we introduce frame interpolation and adapt it to our pipeline to alleviate this issue.
To summarize, **we not just simply combine these techniques but adapt them into the editing of long videos and keep the pipeline training-free.** The techniques are critical enabling tools to improve the quality of the edited videos while ensuring accurate controls.

**Updated ablation study of cross-window attention**:

As shown in Fig 5a), we add experiments comparing cross-window attention to the separate edit of each window based on naive cross-frame attention. We also include fully cross-frame attention-based ControlVideo-I using a hierarchical sampler as a baseline. We find that with our cross window attention can get results comparable to the fully cross-frame attention while taking less attention computation.

**Updated supplementary of comparison videos**:

We add four comparison videos with baselines in [comparison_videos.zip] to visualize the video editing effects. It is better to illustrate the superiority of our method with comparison videos.

---

### Meta-Review · Area_Chair_bzsq · 2023-12-10

**Metareview:**

The manuscript received three reviews which placed it below the acceptance bar. The reviewers had concerns about the novelty, visual and numerical comparisons. They mentioned that the paper is based on ideas proposed earlier, combining them to achieve long video editing. Furthermore, the evaluation is performed on 30 videos only. While authors pointed out that sometimes in the past similar-sized datasets were used for evaluation, reviewer EDK1 believes that this is not sufficient for a paper submitted for a publication in 2024, as no conclusion can be drawn from such a small experiment. The authors provided their detailed response, however, reviewers kept their score. Since the paper didn't receive the necessary support, the AC decides to reject the manuscript.

**Justification For Why Not Higher Score:**

All three reviewers placed the paper below the bar.

**Justification For Why Not Lower Score:**

N/A

---

### Decision · Program_Chairs · 2024-01-16

Reject